# Predicting the Effect of Meropenem Against *Klebsiella pneumoniae* Using Minimum Inhibitory Concentrations Determined at High Inocula

**DOI:** 10.3390/antibiotics14030258

**Published:** 2025-03-03

**Authors:** Maria V. Golikova, Kamilla N. Alieva, Elena N. Strukova, Julia R. Savelieva, Daria A. Kondratieva, Svetlana A. Dovzhenko, Mikhail B. Kobrin, Vladimir A. Ageevets, Alisa A. Avdeeva, Stephen H. Zinner

**Affiliations:** 1Department of Pharmacokinetics & Pharmacodynamics, Gause Institute of New Antibiotics, 11 Bolshaya Pirogovskaya Street, 119021 Moscow, Russia; qvimqwem@yandex.ru (K.N.A.); kindyn@yandex.ru (E.N.S.); savmos80@mail.ru (J.R.S.); goawaymrway@gmail.com (D.A.K.); sad35@yandex.ru (S.A.D.); mbkobrin@gmail.com (M.B.K.); 2Pediatric Research and Clinical Center for Infectious Diseases, 9 Prof. Popov Street, 197022 St. Petersburg, Russia; ageevets@list.ru (V.A.A.); avdeenko-alya@mail.ru (A.A.A.); 3Department of Medicine, Harvard Medical School, Mount Auburn Hospital, 330 Mount Auburn St., Cambridge, MA 02138, USA; szinner@mah.harvard.edu

**Keywords:** minimum inhibitory concentration at high bacterial inocula, antibiotic resistance, carbapenems, OXA-48 carbapenemase, KPC carbapenemase, meropenem, mutant selection window, inoculum effect, *Klebsiella pneumoniae*, in vitro hollow-fiber infection model

## Abstract

**Background/Objectives:** Assessing antibiotic MICs at high bacterial counts is likely to disclose hidden bacterial resistance and the inoculum effect if present and therefore also reveal potential decreased antibiotic effectiveness. In the current study, we evaluated the predictive potential of MICs determined at high bacterial inocula to evaluate meropenem effectiveness and emergence of resistance in *Klebsiella pneumoniae*. **Methods:** Nine carbapenemase-free or carbapenemase-producing *K. pneumoniae* strains were exposed to meropenem in an in vitro hollow-fiber infection model (HFIM). The treatment effects were correlated with simulated antibiotic ratios of the area under the concentration–time curve (AUC) to the MIC (AUC/MIC) and to MICs determined at high inocula (AUC/MIC_HI_). **Results:** Based on MICs determined at standard inocula, meropenem effects at different AUC/MIC ratios for both carbapenemase-free and carbapenemase-producing *K. pneumoniae* strains were stratified and could not be described by a single relationship. In contrast, when AUC/MIC_HI_ ratios were used, a single relationship with the antibiotic effect was obtained for all tested strains. Similarly, the emergence of meropenem resistance in HFIM was concordant with AUC/MIC_HI_, but not with AUC/MIC ratios. **Conclusions:** MICs determined at high bacterial inocula enable the prediction of meropenem effects both for carbapenemase-free and for carbapenemase-producing *K. pneumoniae* strains. Also, MICs at standard and high inocula can identify carbapenemase-producing strains by revealing the inoculum effect.

## 1. Introduction

The clinical choice of antibiotic therapy is based on the in vitro assessment of minimum inhibitory concentrations (MICs), which indicate the potential effectiveness of treatment [1,2,3]. Standard MIC testing is conducted at bacterial concentrations of 5 × 10^5^ CFU/mL. However, during treatment, the emergence of antibiotic-resistant cells within the total bacterial population [4] can limit the predictive potential of “standard” MICs, as they do not necessarily include antibiotic-resistant subpopulations due to the relatively low inoculum. The widespread incursion of antimicrobial-resistant bacteria has prompted the search for other parameters that can evaluate the expected effects of antibiotics on these resistant subpopulations. One such parameter, the mutant prevention concentration (MPC), is assessed at a very high inoculum of 10^10−11^ CFU/mL [5]. The bacterial sample in this method is likely to contain resistant cells, but there are concerns about the routine use and clinical utility of the MPC because its assessment method is complex. This has prompted a search for alternative parameters to more easily and accurately predict antimicrobial resistance.

From a clinical perspective, most severe infections are characterized by high bacterial concentrations at the site of infection, including endocarditis, pneumonia, cystitis and even bacteremia, among others [6,7,8,9]. In this light, assessing an antibiotic’s MIC in the presence of high bacterial concentrations (MIC_HI_) is a logical and possibly essential step to optimally predict the antimicrobial effect. This is particularly important due to the “inoculum effect”, which significantly reduces the activity and effectiveness of β-lactam antibiotics [10]. The clinical relevance of the inoculum effect is still a matter of debate, but several in vivo studies, including clinical trials, have demonstrated its clinical significance [11,12,13,14,15].

In our previous study on meropenem and *Klebsiella pneumoniae* strains that produce carbapenemases, we showed that assessing MICs at a high inoculum allows the accurate evaluation of meropenem efficacy and enrichment of resistant subpopulations [16]. The current investigation aims to extend this research by including strains that exhibit varying susceptibility to meropenem but do not produce carbapenemases. It is important to determine whether assessing MICs for these organisms at high bacterial inocula yields reliable or erroneous results, given that clinical decisions are often based solely on MIC determinations rather than determining carbapenemase production. We also included three carbapenemase-producing strains of *K. pneumoniae* of the KPC type, a significant threat to the efficacy of β-lactam antibiotics [17,18].

To investigate the predictive potential of MICs determined at high bacterial inocula on the antimicrobial effect and enrichment of resistant subpopulations, we assessed meropenem MICs and MIC_HI_s for *K. pneumoniae* strains and analyzed them alongside the results of experiments in a hollow-fiber infection model [19,20]. Using the hollow-fiber infection model, we simulated the pharmacokinetics of meropenem in epithelial lining fluid (ELF). This simulation is clinically relevant given that pneumonia frequently occurs in patients with a weakened immune system, for whom the administration of effective antimicrobial therapy is essential. To optimize antibiotic dosing, it is important to consider pharmacokinetics at the infection site, which might differ considerably from plasma.

## 2. Results

### 2.1. Meropenem MICs at Standard and High Inocula

We have previously shown that assessment of MICs at high bacterial concentrations (MIC_HI_) accurately reflects the efficacy of meropenem against *K. pneumoniae* strains that produce OXA-48 carbapenemases. To generalize our findings, we aimed to supplement these results with data on *K. pneumoniae* strains that do not produce carbapenemases and those that produce a different carbapenemase, KPC. We assessed meropenem MICs at two inocula, standard and 100-fold higher (Table 1).

As seen in the table, there was no difference in MICs at standard or high bacterial densities for all strains that do not produce carbapenemases. This indicates that an increase in bacterial concentration does not affect the activity of meropenem, so there is no inoculum effect. In contrast, all three carbapenemase-producing strains showed an increase in MIC and, hence, an inoculum effect, as expected.

Based on the results for all carbapenemase-non-producing strains, increasing the inoculum did not result in a change in MIC, i.e., it seems that there are no likely hidden risks that the antibiotic might become ineffective at high bacterial concentrations against meropenem-susceptible strains. On the other hand, KPC-producing strains demonstrated significant inoculum-related decreases in meropenem activity, suggesting that its efficacy might be diminished in the presence of a high bacterial concentration. We conducted pharmacodynamic simulations to test these assumptions.

### 2.2. Meropenem Pharmacodynamics with Carbapenemase-Non-Producing K. pneumoniae Strains

The time–kill curves of carbapenemase-non-producing *K. pneumoniae* strains exposed to a clinically relevant meropenem regimen are presented in Figure 1. The meropenem antimicrobial effect was most pronounced against meropenem-susceptible *K. pneumoniae* strains 782 and 2684, with MICs equal to 1 (MIC_HI_ of 2 µg/mL) and 2 µg/mL (MIC_HI_ of 2 µg/mL); the bacteria were fully eradicated and selection of resistance was not observed. Another meropenem-susceptible strain (2286-MIC_HI_ 2 µg/mL) was characterized by a decrease in bacterial concentration: bacterial counts stayed around 4 log(CFU/mL) and the growth of cells resistant to 2× MIC was negligible at the end of the observation. With meropenem-non-susceptible *K. pneumoniae* strains 1676 (MIC_HI_ 4 µg/mL) and 844 (MIC_HI_ 8 µg/mL), the meropenem effect was similar to strain 2286; the selection of resistant mutants was not observed. Only with the meropenem-resistant strain 2895 (MIC_HI_ 16 µg/mL) was meropenem not effective, as bacterial counts, after a slight decrease, reached the initial level of 8 log(CFU/mL) and meropenem resistance (to 2× and 4× MIC) was observed. These results were well predicted by MIC_HI_s.

The coefficients of variation (CVs) for experiments in HFIM ranged from 0.2% to 27% for total populations and from 0.1% to 31.3% for subpopulations resistant to 2–16× MIC of meropenem. However, the CVs for most of the experiments were ≤10%.

### 2.3. Meropenem Pharmacodynamics with Carbapenemase-Producing K. pneumoniae Strains

Meropenem was not effective against carbapenemase-producing strains (Figure 2). Data for *K. pneumoniae* ATCC BAA-1904 from our previous study [16] are presented together with the data for *K. pneumoniae* 6570 and 1714. The intense growth of the total bacterial count and the enrichment of resistance at all levels were observed with all strains. The most intense growth of resistant cells was in experiments with strain 6570, which had a relatively lower MIC of meropenem (4 µg/mL) compared to BAA-1904 (8 µg/mL) and 1714 (32 µg/mL). If we compare the effects of meropenem on bacterial strains with identical MICs (4 and 8 µg/mL) that are either producing (6570 and BAA-1904) or not producing (1676 and 844) carbapenemases, we observe a significant difference in the responses to meropenem, with either no effect or significant effectiveness, respectively. However, when considering the MIC_HI_s in conjunction with the pharmacodynamic data, simulated meropenem treatments were ineffective against those strains that were identified as resistant to meropenem and demonstrated the inoculum effect.

### 2.4. “AUC/MIC-Effect” and “AUC/MIC_HI_-Effect” Relationships

The concordance of MIC and pharmacodynamic data was established between the simulated AUC/MIC or AUC/MIC_HI_ ratios and the antimicrobial effect (Figure 2). In these relationships, AUBC inversely reflects the meropenem effect against *K. pneumoniae*: the higher the effect, the lower the AUBC. It is essential for data to allow a single, unified relationship, which could enable its use in predicting the effect of meropenem whether or not a given strain produces carbapenemase.

As seen in Figure 3a, using MICs at standard inocula, data points for carbapenemase-non-producing and carbapenemase-producing strains were stratified and could not be described by a single relationship. Therefore, at the same AUC/MIC, the predicted effect of meropenem differs whether or not the strain is producing carbapenemases. However, with the use of MIC_HI_s, the data points align so that they can be described by a single relationship with a high *r*^2^ of 0.97 (Figure 3b).

This indicates that the effect of meropenem can be uniformly predicted based on the estimated susceptibility of *K. pneumoniae* strains at high bacterial inocula, regardless of the ability of the strain to produce carbapenemases.

### 2.5. Concordance Between the MIC and MIC_HI_ and Emergence of Resistance

The tested *K. pneumoniae* strains were arranged in order of increasing MICs (Figure 4). As seen in the figure (left panel), there was no concordance between the selection of meropenem resistance and bacterial MICs: strains that are more susceptible to meropenem developed more intense meropenem resistance than less meropenem-resistant strains. In contrast, concordance was observed between MICHIs and AUBCMs for all strains; the highest AUBCMs corresponded to the highest MICHIs for meropenem-resistant strains that demonstrated the inoculum effect.

Thus, for strains that do not produce carbapenemases, the use of MIC_HI_ adequately assesses susceptibility and does not yield false predictions regarding the enrichment of meropenem-resistant subpopulations in response to meropenem exposure. This also holds true for carbapenemase-producing strains, but this does not apply to MICs assessed at the standard inoculum.

## 3. Discussion

The increasing worldwide prevalence of carbapenemase-producing *K. pneumoniae* has exaggerated the need to predict antibiotic efficacy using simple and readily available methods. One such promising method is the estimation of antibiotic MICs at inocula 100-fold higher than the usual standard, as it allows the screening of strains that might be subjected to the inoculum effect. Inoculum-induced MIC elevation is a well-established phenomenon [10,21]. With β-lactams and β-lactamase producing bacteria, the inoculum effect is linked to antibiotic enzymatic degradation resulting from the accumulation of β-lactamases elaborated by a large number of bacteria in a confined space [22]. However, it is not sufficient to simply evaluate antibiotic MICs at different bacterial concentrations; it is also necessary to understand if inoculum-induced MIC increases affect treatment efficacy. Some opinions suggest that the inoculum effect is a laboratory artifact of little clinical significance [23]. Studies in mice have shown that the inoculum effect plays a role in antibiotic responses to bloodstream [11], peritoneal [12] and lung [24] infections. Clinically, the inoculum effect was identified in cefazolin-treated patients with bacteremia due to β-lactamase-producing, methicillin-susceptible *Staphylococcus aureus* [13,14]. The verification of the inoculum effect can also be demonstrated in in vitro conditions using the hollow-fiber infection model, which can assess and predict the efficacy of antibiotic treatments as well as determine optimal dosing strategies [19,20]. Using such a model, the significance of the inoculum effect with Group A Beta-hemolytic *Streptococci* was demonstrated for linezolid, clindamycin and benzylpenicillin [25,26].

In our previous study, we evaluated meropenem against OXA-48 carbapenemase-producing *K. pneumoniae* strains that included meropenem-susceptible variants [16]. The results showed that assessing meropenem MICs at high bacterial concentrations can identify strains assessed as “susceptible” using the traditional method that have a high risk of treatment failure as observed in pharmacodynamic simulations. The goal of the current study was to broaden the range of strains studied and include those that do not produce carbapenemases and those that produce clinically important KPC-type carbapenemases [17,18].

The inclusion of strains that do not produce carbapenemases allows for the evaluation of MICs determined at a high bacterial concentration as a reliable method of susceptibility assessment, independent of carbapenemase production by the strain.

Figure 3 and Figure 4 show that the effect of meropenem and the enrichment of resistant cells can be adequately predicted using MICs determined at high inocula. With carbapenemase-non-producers, the inoculum effect was not observed and MIC_HI_s were equal to standard MICs. However, with KPC carbapenemase producers, MIC_HI_s differed significantly from standard MICs due to the inoculum effect. Obviously, the standard MIC assessment does not identify these differences. Similarly, the inoculum effect was observed for all OXA-48 carbapenemase-producing strains, as reported in our previous study [16]. To summarize the results obtained in both our current and previous studies, we added the data on OXA-48-producing strains to Figure 3b (Figure 5). As seen in Figure 5, the merged data were arranged in a manner such that they can be described by a single sigmoid relationship with a high r-squared correlation coefficient (0.97). This further indicates that MIC_HI_s can predict the effect of meropenem on strains with varying resistance patterns.

We have further analyzed combined data on the selection of meropenem resistance in tested strains and their MICs, obtained from the current and the companion study and designed Figure 6. As seen in the figure, when MIC was assessed at high bacterial load (MIC_HI_), the selection of resistance corresponded to the assessment of the strain as resistant, regardless of whether or not it produced carbapenemase. In contrast, when the MIC was measured using the standard method, there was no concordance with the selection of resistance observed in pharmacodynamic studies.

According to our results, high bacterial counts in pharmacodynamic experiments led to the enrichment of resistant subpopulations that most likely survived meropenem exposure by actively producing β-lactamases. The increased expression of carbapenemase genes in the response to antibiotic pressure was previously described as a bacterial survival strategy [27]. In addition, it has been demonstrated that the enrichment of meropenem-resistant cells among OXA-48 carbapenemase-producing *K. pneumoniae* strains [28] follows a concept known as the “mutant selection window” [29], which has been previously described for other antibiotic classes that are unrelated to beta-lactams [30,31,32]. In one of these studies, the most intense selection of resistant cells was observed not in the most resistant strains of *K. pneumoniae* with MICs of 8–32 µg/mL, but rather in strains with intermediate MIC values of 2 and 4 µg/mL [28].

This occurred at the peak of a bell-shaped relationship described by a Gaussian function. In contrast, more or less resistant strains (with MICs of 0.5 or 8–32 µg/mL) showed either a less intense selection of resistance or no selection at all. In the current study, we also observed that the most intense growth of resistant cells in the pharmacodynamic experiments occurred with strain 6570, which had a relatively lower MIC value of 4 µg/mL compared to the other two strains (MICs of 8 and 32 µg/mL). This can be explained by the concept of the “mutant selection window”, where the strain with MIC of 4 µg/mL appears at the apex of a bell-shaped curve, but strains with higher MICs apepar on its descending branch. It would be useful to investigate the correlation between resistance selection and meropenem exposure to evaluate the applicability of the “mutant selection window” for KPC-producing *K. pneumoniae*. However, we did not obtain strains with meropenem MICs lower than 4 µg/mL, probably because KPC carbapenemase enzymes typically confer a high degree of resistance to meropenem [33]. However, to observe such a relationship among the studied strains, we conducted additional data analysis. We explored the connection between the selection of meropenem resistance and the strain MIC (as the denominator in the AUC/MIC or the AUC/MIC_HI_ ratio) and validated the applicability of the MSW concept for *K. pneumoniae* and meropenem. To obtain adequate data points for analysis, we merged the data for OXA-48 and KPC carbapenemase-producing strains, as well as those that do not produce enzymes. This investigation yielded interesting findings (see Figure 7). As seen in the figure, when conventional MIC values were used to calculate AUC/MIC ratios, no correlation was observed. However, when MIC_HI_s were used, a relationship between resistance selection and AUC/MIC_HI_ values could be described with a Gaussian function with a high degree of correlation. This means that MIC_HI_s has a potential as an alternative to the MPC as the upper border of the MSW. Consequently, MIC_HI_s may be useful in predicting meropenem resistance in *K. pneumoniae*. According to our analysis, strains with meropenem MIC_HI_s of 64 and 128 µg/mL are the most concerning in the selection of meropenem resistance, and strains with MIC_HI_ of 8 µg/mL and lower are less concerning.

In fact, among strains of *K. pneumoniae* resistant to meropenem, there are variants with threshold MIC levels of 16 µg/mL. These variants are not rare and often circulate in hospitals (numerous studies have confirmed this [34,35]). Moreover, strains with lower MIC values, such as those that are non-susceptible to meropenem with MICs of 4 or 8 µg/mL, or those that are susceptible to an MIC of 2 µg/mL, are of particular concern. The reason is that, by solely focusing on MIC values, the risk of ineffective meropenem treatment may be underestimated, as carbapenemases can lead to treatment failure. The phenomenon of phenotypical susceptibility to carbapenems in carbapenemase-producing Enterobacteriaceae has been described previously [36]. Unfortunately, there is no confirmation that the MIC will increase with the ability of the strain to produce beta-lactamases.

It might be beneficial to introduce an evaluation of MIC at high bacterial loads in clinical laboratories. From a technical perspective, this is even easier than standard MIC assessment, as it does not require the dilution of a bacterial suspension from a density of 10^8^ to 10^6^ CFU/mL to determine the MIC. However, variability in MIC determinations can indeed affect the assessment of MIC, MIC_HI_ and the inoculum effect, the determination of which is critical to reveal meropenem-susceptible strains that pose a hidden threat. It will be necessary to assess the variability of MIC estimations in larger bacterial samples to ensure their accuracy and diminish risk of false results.

A limitation of our study is the use of only one antibiotic. To more reliably assess antibiotic efficacy based on MIC_HI_, it would be necessary to evaluate additional antibiotics and bacteria, including those with various resistance mechanisms, such as the ability to produce NDM-carbapenemases. Additionally, more dosing regimens and antibiotic exposures at various sites of infection should be simulated. Since KPC-producing strains have demonstrated high MIC_HI_ (all producers are resistant to meropenem) due to the presence of carbapenemases, other mechanisms of resistance may be considered negligible, so we did not study them.

## 4. Materials and Methods

### 4.1. Antimicrobial Agent and Bacterial Isolates

Meropenem powder was purchased from Sigma-Aldrich (St. Louis, MO, USA). Six carbapenemase-non-producing clinical isolates of *K. pneumoniae* collected from clinical samples of ICU patients admitted to Moscow and Saint-Petersburg hospitals were used in the study: 782, 2684, 2286, 1676, 844 and 2895. Two KPC carbapenemase-producing (by PCR) clinical isolates of *K. pneumoniae* collected from clinical samples of ICU patients admitted to Moscow and Smolensk hospitals were used in the study: 6570 and 1714. The final choice was based on strain meropenem susceptibility in order to identify strains with a wide range of meropenem susceptibility that includes meropenem susceptible, non-susceptible and resistant strains. Additionally, we were interested in the carbapenemases produced by these strains; we focused on OXA-48- and KPC-producing strains, as these carbapenemases are widely distributed worldwide. To select for such strains, all strains were screened for the presence of the most common carbapenemase genes, *bla*KPC, *bla*OXA-48-like, *bla*NDM, *bla*IMP and *bla*VIM, using polymerase chain reaction (PCR) with the previously described specific primers and reaction conditions [37].

### 4.2. Susceptibility Testing

MICs were determined by the broth microdilution technique at an inoculum size of 5 × 10^5^ CFU/mL in Mueller–Hinton broth (MHB) (Becton Dickinson, Holdrege, NE, USA) at an inoculum size of 5 × 10^5^ CFU/mL [38]. Additionally, to see whether inoculum size affects MIC values, MICs at a high inoculum of 5 × 10^7^ CFU/mL (MIC_HI_s) were evaluated by the broth microdilution technique. Susceptibility testing was repeated at least three times each. The MIC breakpoint for meropenem susceptibility was used according to EUCAST recommendations [2]. The interpretive criteria for susceptibility were as follows: susceptible, ≤2 µg/mL; resistant, >8 µg/mL. *K. pneumoniae* ATCC 700,603 served as a reference control in MIC testing [39].

### 4.3. In Vitro Dynamic Model and Operational Procedure Used in the Pharmacodynamic Experiments

A previously described two-compartment in vitro model (a hollow-fiber infection model) [40] was used in pharmacodynamic simulations with meropenem. Briefly, the model consists of three connected chambers: one containing fresh cation-supplemented Mueller–Hinton broth (CSMHB), which supplied CSMHB to the second chamber-the central unit used for drug dosing—and the third chamber—a hollow-fiber bioreactor (FiberCell Systems, New Market, MD, USA, cellulose cartridge C3001)—was a peripheral unit used for bacterial cultivation, representing the infection site. The central unit and bioreactor were connected, and the continuous exchange of CSMHB between these units by peristaltic pump (Cole-Parmer Instrument Company, Masterflex L/S 07523-80, Vernon Hills, IL, USA) enabled the maintenance of target drug concentrations in both cameras.

The operational procedure used in the pharmacodynamic experiments was as described in detail elsewhere [40]. Briefly, the system was filled with sterile MHB and placed in an incubator at 37 °C. Each experiment was performed at least in duplicate. Antibiotic dosing and sampling were processed automatically using computer-assisted controls. The system was filled with sterile CSMHB and placed in an incubator at 37 °C. Then, 18 h culture of *K. pneumoniae* was injected into the hollow-fiber bioreactor to produce a starting bacterial concentration ~10^7^ CFU/mL. After a 2 h period of incubation, the resulting exponentially growing bacteria reached ~5 × 10^8^ colony-forming units (CFU)/mL. Then, the antibiotic was administered into the central unit of the model. The duration of each experiment was equal to 120 h.

### 4.4. Antibiotic Dosing Regimens and Simulated Pharmacokinetic Profiles

Meropenem treatment mimicked the therapeutic dosing regimen: 2 g administered every 8 h, as a 3 h intravenous infusion. A mono-exponential profile in human epithelial lining fluid after a thrice-daily dosing of meropenem with a half-life of 1.4 h was simulated for 5 consecutive days [41]. The steady-state CMAX = 32.4 µg/mL and the steady-state 24 h area under the concentration–time curve (AUC) = 375 (mg × h)/L.

The reliability of pharmacokinetic simulations has been verified previously [28]; the thermal degradation of the antibiotic is moderate, and does not significantly distort the expected concentrations of meropenem over the dosing interval.

### 4.5. Quantitation of the Antimicrobial Effect

The bacteria-containing medium from the hollow-fiber bioreactor was sampled to determine bacterial concentrations throughout the observation period. For bacterial enumeration, the samples (100 μL) were serially diluted as appropriate and 100 μL was plated onto tryptic soy agar (Becton Dickinson, USA) plates, which was incubated at 37 °C for 24 h. The lower limit of accurate detection was 2 log CFU/mL (equivalent to 10 colonies per plate).

To determine the time course of meropenem-resistant bacterial concentrations, the samples were plated on Mueller–Hinton agar (Becton Dickinson, USA) plates with a meropenem concentration equal to 2×, 4×, 8× and 16× MIC. The inoculated plates were incubated for 24–48 h at 37 °C and screened visually for growth. The lower limit of detection was 1 log CFU/mL (equivalent to at least one colony per plate).

To quantify the antimicrobial effect of meropenem (from 0 to 120 h), the integral parameters of the area under the time–kill curve (AUBC) and the AUBC for meropenem-resistant organisms (AUBC_M_) were calculated [42]. The greater the antibacterial effect and resistance prevention, the lower the AUBC and AUBC_M_, respectively.

### 4.6. Statistical Analysis

The reported MIC data were obtained by a calculation of the respective modal values. In pharmacodynamic experiments, bacterial count data were calculated as arithmetic mean ± standard deviations for two or three replicate experiments. Based on these data, kinetic growth and time–kill curves were constructed. To facilitate the viewing of the figures, we did not place data point error bars in order to not interfere with the kinetic curves. The coefficient of variation (CV) for the log CFU/mL data of the total population and resistant subpopulations was calculated.

The AUBC versus AUC/MIC and AUC/MIC_HI_ data were fitted by the following sigmoid function:Y = Y0 + a/{1 + exp[−(x − x0)/b]}(1)
where Y is AUBC, Y0 is the maximal Y value, x is log AUC/MIC or AUC/MIC_HI_, a is the maximal value of the antimicrobial effect, x0 is x corresponding to a/2 and b is a parameter reflecting sigmoidicity.

The AUBC_M_ versus AUC/MIC and AUC/MIC_HI_ data were fitted by the following Gaussian function:Y = a × exp{−0.5 × [(x − x_0_)/b]^2^}(2)
where Y is ABBC_M_, x is log AUC/MIC or log AUC/MIC_HI_, x_0_ is log AUC/MIC or log AUC/MIC_HI_ that corresponds to the maximal value of Y and a and b are parameters.

All calculations were performed using SigmaPlot 12 software (Systat Software Inc., headquartered in San Jose, CA, USA).

## 5. Conclusions

Our findings suggest that standard MIC estimations may underestimate the predicted effectiveness of carbapenems against carbapenemase-producing bacteria. Higher bacterial concentrations during MIC testing may enhance the contribution of resistant subpopulations to the MIC estimate under such circumstances. Together, these findings suggest that the implementation of MIC testing at high bacterial concentrations in clinical practice might more rapidly, accurately and cost-effectively identify carbapenemase-producing strains. This, in turn, has the potential to optimize treatment outcomes for patients with infections caused by *K. pneumoniae*.

## Figures and Tables

**Figure 1 antibiotics-14-00258-f001:**
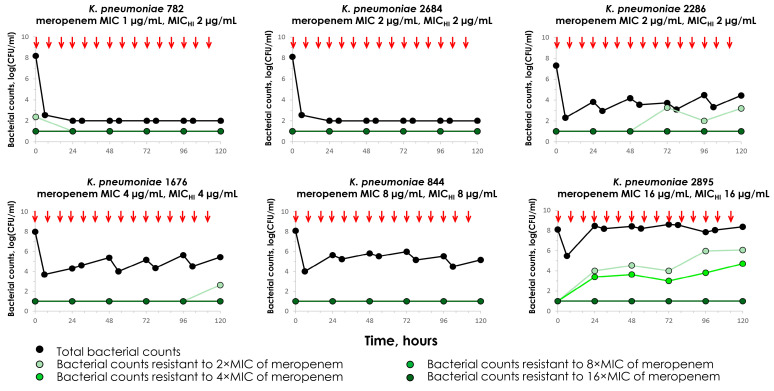
Total and meropenem-resistant (to 2×, 4×, 8× and 16× MIC of meropenem) counts of *K. pneumoniae* in in vitro pharmacodynamic simulations with meropenem against carbapenemase-non-producing strains. Simulated pharmacokinetics: 2 g as a 3 h infusion every 8 h for 5 days in ELF. Arrows indicate the start of meropenem infusions.

**Figure 2 antibiotics-14-00258-f002:**
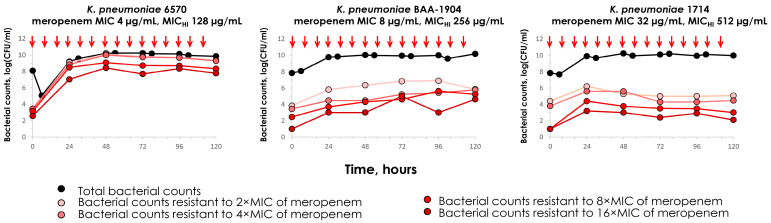
Total and meropenem-resistant (to 2×, 4×, 8× and 16× MIC of meropenem) counts of *K. pneumoniae* in in vitro pharmacodynamic simulations with meropenem against carbapenemase-producing strains. Simulated pharmacokinetics: 2 g as a 3 h infusion every 8 h for 5 days in ELF. The data for *K. pneumoniae* ATCC BAA-1904 strain are reproduced from Alieva et al. [16]. Arrows indicate the start of meropenem infusions.

**Figure 3 antibiotics-14-00258-f003:**
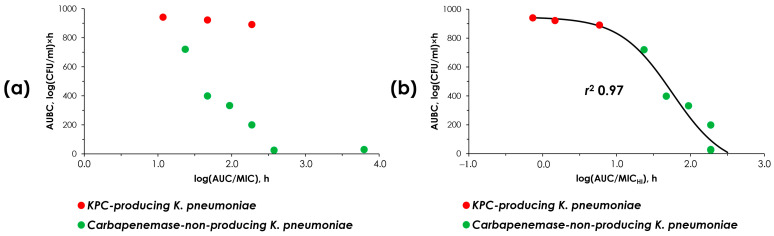
“AUC/MIC-AUBC” and “AUC/MIC_HI_-AUBC” relationships. (**a**) The data assembled using standard MIC values; (**b**) the data assembled using high inocula MIC values (MIC_HI_) and described with the sigmoid function. The data for *K. pneumoniae* ATCC BAA-1904 strain are reproduced from Alieva et al. [16].

**Figure 4 antibiotics-14-00258-f004:**
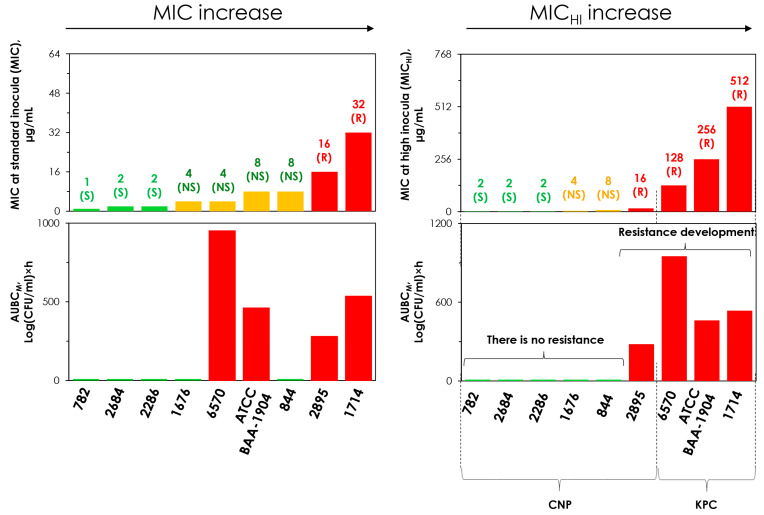
MICs at standard (MIC) and high (MIC_HI_) inocula and the corresponding AUBC_M_s (integral measure of meropenem resistance and area under the bacterial curve resistant to 4× MIC of meropenem, obtained from pharmacodynamic simulations with respective *K. pneumoniae* strains and meropenem). The higher the AUBC_M_, the more intense selection of resistance to meropenem. S, susceptible; NS, non-susceptible; R, resistant. CNP—carbapenemase-non-producing *K. pneumoniae* strains; KPC—KPC-producing *K. pneumoniae* strains. The data for *K. pneumoniae* ATCC BAA-1904 are reproduced from Alieva et al. [16].

**Figure 5 antibiotics-14-00258-f005:**
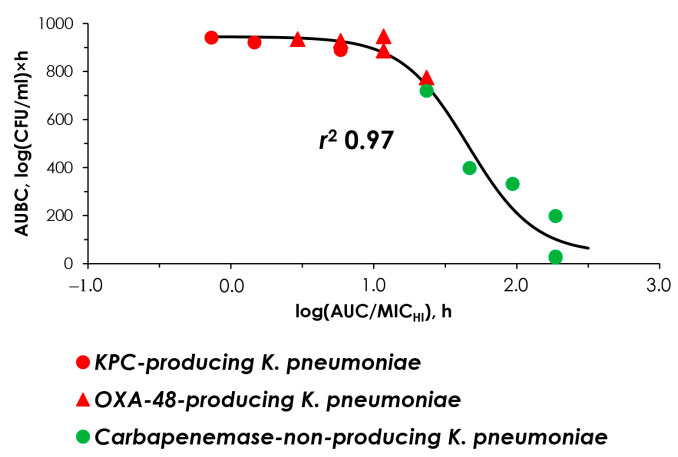
“AUC/MIC_HI_-AUBC” relationship described with the sigmoid function. The data for OXA-48-producing *K. pneumoniae* 1128, 1456, 1170 and KPC-producing ATCC BAA-1904 strains are reproduced from Alieva et al. [16].

**Figure 6 antibiotics-14-00258-f006:**
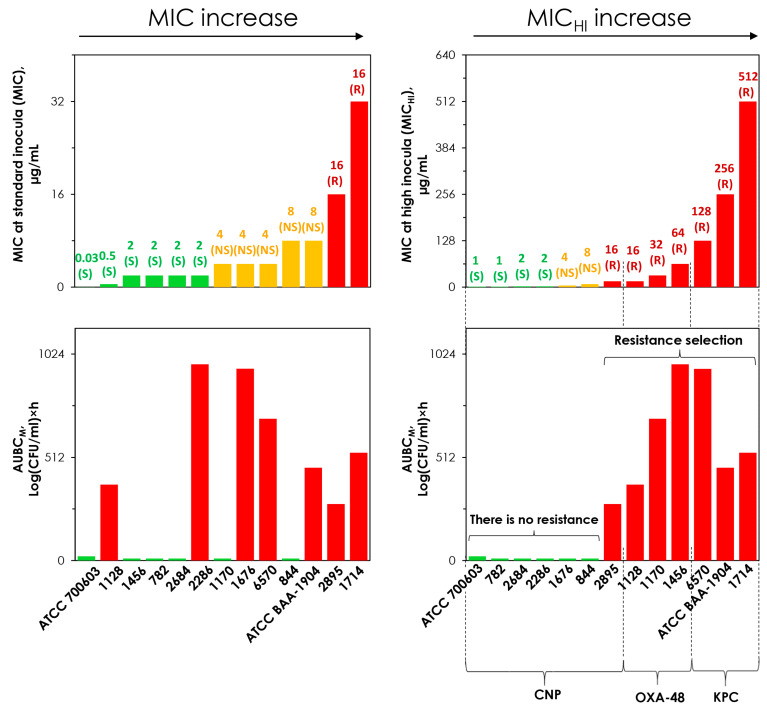
MICs at standard (MIC) and high (MIC_HI_) inocula and the corresponding AUBC_M_s (integral measure of meropenem resistance and area under the bacterial curve resistant to 4× MIC of meropenem, obtained from pharmacodynamic simulations with respective *K. pneumoniae* strains and meropenem). The higher the AUBC_M_, the more intense the selection of meropenem resistance. S, susceptible; NS, non-susceptible; R, resistant. CNP—carbapenemase-non-producing *K. pneumoniae* strains; OXA-48—OXA-48 carbapenemase-producing *K. pneumoniae* strains; KPC—KPC-producing *K. pneumoniae* strains. The data for *K. pneumoniae* ATCC BAA-1904 and all OXA-48-producing strains are reproduced from Alieva et al. [16].

**Figure 7 antibiotics-14-00258-f007:**
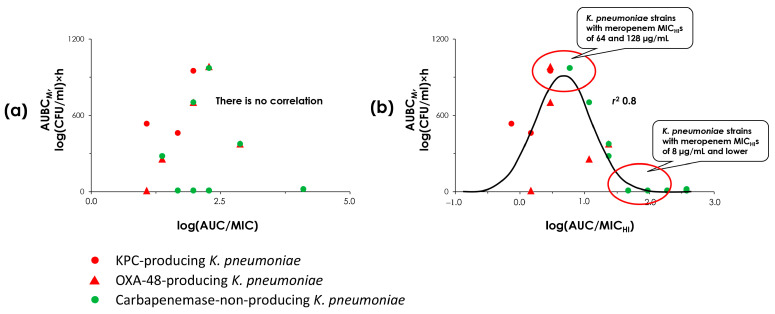
“AUC/MIC-AUBC_M_” and “AUC/MIC_HI_-AUBC_M_” relationships. (**a**) Data assembled using standard MIC values; (**b**) data assembled using high inocula MIC values (MIC_HI_). Data for *K. pneumoniae* ATCC BAA-1904 strain and all OXA-48-producing strains are reproduced from Alieva et al. [16].

**Table 1 antibiotics-14-00258-t001:** Meropenem MICs at standard and high inocula.

№	*K. pneumoniae* Strain	Carbapenemases	Meropenem MIC, µg/mL	Meropenem MIC_HI_, µg/mL	MIC_HI_/MIC Ratio
1	782	None	1 (S)	2 (S)	2
2	2684	None	2 (S)	2 (S)	1
3	2286	None	2 (S)	2 (S)	1
4	1676	None	4 (NS)	4 (NS)	1
5	844	None	8 (NS)	8 (NS)	1
6	2895	None	16 (R)	16 (R)	1
**7**	**6570**	**KPC**	**4 (NS)**	**128 (R)**	**32**
**8**	**ATCC BAA-1904**	**KPC**	**8 (NS)**	**256 (R)**	**64**
**9**	**1714**	**KPC**	**32 (R)**	**512 (R)**	**16**

The strains with an inoculum effect are highlighted in **bold**. R—resistant (MIC > 8 µg/mL); NS—non-susceptible (2 < MIC < 8 µg/mL); S—susceptible (MIC ≤ 2 µg/mL) [2].

## Data Availability

Data are contained within the article.

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
