# Peer review of "Predicting the Effect of Meropenem Against Klebsiella pneumoniae Using Minimum Inhibitory Concentrations Determined at High Inocula"

_antibiotics, 2025, doi:10.3390/antibiotics14030258_

Round 1
Reviewer 1 Report
Comments and Suggestions for Authors
1. Could you clarify how the current study expands on previous findings with OXA-48-producing strains? Is the inclusion of KPC-producing strains intended to confirm the universality of the MICHI approach?
2. Is the choice of only one antibiotic (meropenem) sufficient to generalize the utility of MICHI across carbapenem-resistant strains? Are there plans to expand this approach to other β-lactam antibiotics?
3. Given the inoculum effect's significant impact on β-lactam antibiotics, is it advisable to propose MICHI testing as a standard clinical practice?
4. Can the you elaborate on how the bacterial strains were selected? Were these representative of diverse clinical scenarios across institutions?
5. Were the simulated conditions in the hollow-fiber model reflective of all relevant clinical infection sites (e.g., bloodstream, lung tissue)? Could this limitation affect the generalizability of findings?
6. Could you provide more technical details on reproducibility when determining MICHI? Were there inter-experiment variations that might limit its reliability?
7. The mutant selection window concept is discussed briefly. Could you provide further analysis or examples of how MICHI aligns with this framework in resistant subpopulation dynamics?
8. The study presents AUC/MICHI relationships with a high r-squared value. Can you explain potential outliers or conditions under which this relationship may break down?
9. For KPC-producing strains, were there any unexpected patterns of resistance enrichment at intermediate MICHI values? Could these suggest previously unknown resistance mechanisms?
Reviewer 2 Report
Comments and Suggestions for Authors
PrediThe manuscript authored by Golikova et al., entitled Predicting the effect of Meropenem against Klebsiella pneumoniae using MICs determined at high inocula, investigates the evolution of antimicrobial resistance and antibacterial effects of Meropenem against high bacterial counts. The manuscript requires significant revision before being considered for publication. The following are the major point to revise:
Authors are requested to furnish the high-resolution image of Figure 1, and improvements in the figure legends and axes are required.
The authors also showed a graph of minimum inhibitory concentrations and MICs at high inocula in Figure 3. No MIC curve is provided, which is essential for publication. Moreover, since the Klebsiella pneumoniae isolate is resistant, one would expect the MIC to be much higher than reported. The authors should explain why that is. A non-resistant Klebsiella pneumoniae isolate can be used for comparison with drugs to which resistant Klebsiella pneumoniae is sensitive.
The effect of drugs should also be clearly shown by a CFU assay using multiple drug concentrations after treatment. A more thorough understanding of the effect on Klebsiella pneumoniae needs to be seen.
The authors have found the drug's efficacy, if the genetics/ phenotype of this isolate has not been tested previously, please mention the references. They checked the antibiotic susceptibility pattern of the isolate against only the Meropenem drug but without any control. Generally, a wild-type strain is used as a control for testing any drug with another clinical isolate that has been tested rigorously.
Please mention whether the susceptibility test was done at least in triplicate.
Please mention the reference or the methods in detail you followed for analysis in the material method section.
Authors are requested to improve the result and discussion section and duly incorporate all the recommendations in the revised manuscript.
he effect of meropenem against
Klebsiella pneu- 2
moniae
using MICs determined at high inoculaPredicting the effect of meropenem against
Klebsiella pneu- 2
moniae
using MICs determined at high inocula Predicting the effect of meropenem against
Klebsiella pneu- 2
moniae
using MICs determined at high inoculaPredicting the effect of meropenem against
Klebsiella pneu- 2
moniae using MICs determined at high inocula
Minor grammatical errors and formatting are required.
Round 2
Reviewer 1 Report
Comments and Suggestions for Authors
The article has been improved, and the authors have satisfactorily answered all questions, so I propose the article for publication.